# Sequencing and Structural Analysis of the Complete Chloroplast Genome of the Medicinal Plant *Lycium*
*chinense* Mill

**DOI:** 10.3390/plants8040087

**Published:** 2019-04-03

**Authors:** Zerui Yang, Yuying Huang, Wenli An, Xiasheng Zheng, Song Huang, Lingling Liang

**Affiliations:** 1DNA Barcoding Laboratory for TCM Authentication, Mathematical Engineering Academy of Chinese Medicine, Guangzhou University of Chinese Medicine, Guangzhou 510006, China; yzr_gzucm1991@163.com (Z.Y.); 13202078270@163.com (Y.H.); 18434376623@163.com (W.A.); xszheng@gzucm.edu.cn (X.Z.); 2Pharmaceutical School, YouJiang Medical University for Nationalities, Baise 533000, China

**Keywords:** *Lycium chinense* Mill, chloroplast genome, phylogenetic analysis

## Abstract

*Lycium chinense* Mill, an important Chinese herbal medicine, is widely used as a dietary supplement and food. Here the chloroplast (CP) genome of *L. chinense* was sequenced and analyzed, revealing a size of 155,756 bp and with a 37.8% GC content. The *L. chinense* CP genome comprises a large single copy region (LSC) of 86,595 bp and a small single copy region (SSC) of 18,209 bp, and two inverted repeat regions (IRa and IRb) of 25,476 bp separated by the single copy regions. The genome encodes 114 genes, 16 of which are duplicated. Most of the 85 protein-coding genes (CDS) had standard ATG start codons, while 3 genes including *rps12*, *psbL* and *ndhD* had abnormal start codons (ACT and ACG). In addition, a strong A/T bias was found in the majority of simple sequence repeats (SSRs) detected in the CP genome. Analysis of the phylogenetic relationships among 16 species revealed that *L. chinense* is a sister taxon to *Lycium barbarum*. Overall, the complete sequence and annotation of the *L. chinense* CP genome provides valuable genetic information to facilitate precise understanding of the taxonomy, species and phylogenetic evolution of the Solanaceae family.

## 1. Introduction

*Lycium chinense* plays an important role in Chinese traditional medicine and has been used as a food and herbal medicine in China for thousands of years [1,2]. The fruits of *L. chinense* have potential pharmacological effects such as anti-aging, reducing blood glucose and serum lipids, immune regulation, among others [3]. Moreover, the dry root bark of *L. chinense* is widely used for the treatment of night sweats, diabetes, coughs, vomiting blood, high blood pressure, and ulcers, and is officially listed in the Chinese Pharmacopoeia (2015 version) [4].

*L. chinense* belongs to the Solanaceae family, which is composed of around 27,000 species belonging to 24 genera. Among them, *Lycium* is one of the most important genus because the fruit, leaf, root bark, and young shoots of many species of this genus have been used as local foods and/or medicines for many years [5]. The *Lycium* genus comprises approximately 70 species, which are widely distributed throughout the world, including southern Africa, Europe, Asia, America and Australia [6,7], however, seven are unique to China. These seven Chinese species are all deciduous shrubs with highly similar morphologies and structures, making it difficult to distinguish among them according to their appearance. Consequently, they are often confused in the market [8]. There has been an attempt to define DNA barcode sequences to identify and analyze the phylogenetic relationships within the *Lycium* genus, however, this has been unsuccessful [8]. Therefore, more effective molecular markers are required to facilitate the investigation of the relationships within the *Lycium* genus [9].

In recent years, chloroplast (CP) genomes have been widely used to reconstruct phylogenetic relationships among various land plants. As an important plastid, CPs play an indispensable role in a plant cell’s utilization of photosynthesis, carbon fixation. With the rapid development of next-generation sequencing technologies, sequencing of entire genomes is feasible for most laboratories [10]. According to published reports, the structure of CP genomes in angiosperms are highly conserved, with an approximate length of 150 kb comprised of two inverted repeat regions (IRa and, IRb) and large and small single copy regions (LSC and SSC), with the two inverted repeats (IRs) separated by the two single copy regions (SCs) [11,12]. CP genomes can be used to study evolutionary relationships at the taxonomic level, as they are maternally inherited, haploid, and highly conserved in gene content and genome structure [13,14].

Transcriptome analysis of *L. chinense* leaves has been reported [15], however, information about the structure of its CP genome is still lacking. Here we report the CP genome of *L. chinense* based on data generated using next-generation sequencing. Further, gene structure characteristics, RNA editing sites, codon usage as well as phylogenetic position of *L. chinense* were analyzed and compared with those of several related species. To the best of our knowledge, this is the third comprehensive analysis of a CP genome from the *Lycium* genus, along with the two recently reported CP genomes from *Lycium barbarum* and *Lycium ruthenicum*.

## 2. Results

### 2.1. CP Genome Organization and Gene Content

The CP genome of *L. chinense* is 155,756 bp in length consisting of large and small single-copy regions of length 86,595 and 18,209 bp, separated by two IR regions of 25,476 bp (Figure 1 and Table 1). The total G + C content was similar to that of other species in the Solanaceae family, which is about 37.8% [11,12,16,17]. And the G + C contents of the LSC (35.8%) and SSC regions (32.3%) were lower than those of the IR regions (43.1%). In the protein-coding regions (CDS), the G + C contents of the first, second and third codon positions were 43.9%, 37.9%, and 33.1%, respectively (Table 1), demonstrating that the A + T content is much higher at the third codon position than those of the other two. This phenomenon, which is commonly detected in other plant CP genomes, can be used to separate nuclear and mitochondrial DNA from CP DNA [18,19,20,21]. 

The *L. chinense* CP genome is predicted to encode a total of 130 functional genes. Among them, 16 are duplicated in the IR region, including 5 CDS, 7 tRNAs and 4 rRNAs. Annotation revealed 80 distinct protein-coding genes, 30 separate tRNA genes and 4 distinct rRNA genes (Table 2). In addition, one gene (*ycf1*) without a stop codon was annotated as a pseudogene. Intron-containing genes were also analyzed. In total, there were 17 genes containing one or two introns found in the *L. chinense* CP genome (Table 3), including 12 CDS and 5 *tRNA* genes, two of which (*ycf3* and *clpP)* contain two intron. While there was a single intron in the other 15 genes, as has also been reported in other plants [22,23].

### 2.2. Repeat Structure and Simple Sequence Repeat (SSR) Analysis 

The SSRs detected in *L. chinense* are represented in Table 4. A total of 231 SSR loci were detected, including 107 mononucleotide, 46 dinucleotide, 67 trinucleotide and 11 tetranucleotide repeat units. In addition, of the mononucleotide SSRs, 99.1% constituted A/T sequences, while only one belongs to G/C motif. Interestingly, 63.0% of the dinucleotide SSRs were also A/T motifs. 

Long repeats are defined as sequence repeats ≥30 bp, which might function to increase population genetic diversity and promote chloroplast genome rearrangement [24]. It must be noted that the repeat types found are all forward and palindromic among the four species which all belong to the Solanaceae family. There are 49 (25 forward, 24 palindromic), 46 (23 forward, 23 palindromic), 39 (20 forward, 19 palindromic), 40 (20 forward, 20 palindromic) large repeats in the CP genomes of *L. chinense*, *L. barbarum*, *L. ruthenicum*, *A. belladonna*, respectively (Figure 2). 

### 2.3. Comparative Chloroplast Genomic Analysis

To facilitate subsequent phylogenetic analyses and plant identification smoothly, we used the mVISTA program to analyze the entire CP genome sequence of *L. chinense* in comparison with those of *A. belladonna* (NC_004561.1), *L. barbarum* (MH032560.1), *L. ruthenicum* (NC_039651.1) (Figure 3). These four species all belong to the Solanaceae family. The data presented in Figure 3 clearly demonstrate that the IR regions are more conserved than the SC regions. Copy correction caused by gene conversion between the two IR region sequences could be the main cause of this phenomenon [25]. Moreover, coding regions were more strongly conserved than the non-coding regions, which is a very common finding in the CP genomes of many other angiosperms [26,27,28].

### 2.4. IR Contraction and Expansion in the L. chinense. CP Genome.

As shown in Figure 4, the IR-SSC and IR-LSC boundaries of *L. chinense* were compared with those of three other species, *A. belladonna* (NC_004561.1), *L. barbarum* (MH032560.1) and *L. ruthenicum* (NC_039651). The length of the IR regions in the four CP genomes ranged from 25,395 to 25,906 bp, demonstrating a modest expansion. Due to the expansion, the *rps19* and *ycf1* genes were partially included in the IR regions of the Solanoideae family. Consequently, there is a truncated *rps19* pseudogene and a *ycf1* pseudogene copy found at the junction of LSC/IRB and SSC/IRA, respectively. The *ndhF* and *trnH* genes, which are entirely located in the SSC and LSC regions, respectively, were at varying distances from the LSC/SSC borders within the Solanoideae family; the longest distance (54 bp) was observed in the *L. chinense*. In contrast, 33 bp of the *rps19* gene and 979 bp of the *ycf1* gene are extended into the IR regions in the *L. ruthenicum* CP genome, which is the shortest among the four species. Moreover, in *L. chinense, L. barbarum*, and *A. belladonna*, 47, 49 and 60 bp of the *rps19* gene and 995, 998 and 1438 bp of the *ycf1* gene are extended into the IR regions, respectively. Taken together, these data indicate that the expansion and contraction of the IR/SC regions exhibit similar patterns within the family, with slight variations.

### 2.5. Codon Usage and RNA Editing Sites

The results of analysis of codon usage in the *L. chinense* CP genome are summarized in Figure 5 and Appendix A. Twenty amino acids that can be transported for protein biosynthesis by tRNA molecules were found in the CP genome. All the CDS in the *L. chinense* CP genome combined included 26,569 codons. Among them, codons encoding leucine was the most common, accounting for 13.16% of total usage. While at the same time, codons encoding cysteine were the least frequent, accounting for 1.82% of total usage. Furthermore, as the number of codons encoding a particular amino acid increased, the relative synonymous codon usage (RSCU) value also increased (Figure 5). Interestingly, most amino acid codons, other than two non-redundant codons (for methionine and tryptophan), exhibit preferential use, as observed in other species [29,30,31,32,33].

The Predictive RNA Editor for Plants (PREP) suite was used to detect potential RNA editing sites in the *L. chinense* CP genome, and the results were summarized in Appendix A. A total of 50 RNA editing sites in 35 genes were identified. All the nucleotide changes found were cytidine (C) to thymine (T) edits, which are typical of the transcripts of land plant CP genomes [34]. Furthermore, the results also revealed that the conversion from the amino acid S to L is the most frequent, accounting for approximately 40%; while H to Y, and I to F switches were the least common, accounting for only 2%.

### 2.6. Phylogenetic Analysis

The results of phylogenetic analyses are presented in Figure 6 and Figure 7 and Appendix A; most nodes were strongly supported with bootstrap values of 100%. Furthermore, all 16 species included in this analysis were split into two clades, with 14 Solanaceae plants comprising a unique clade, and the rest two outgroup species from the Labiatae grouped into the other clade. Within the Solanaceae, each genus basically clustered alone, demonstrating good monophyletic separation of this family. *L. chinense* (*Lycium*) was a sister species to *L. barbarum*, which is strongly supported by both the maximum likelihood and BEAST trees. Further, the *Lycium* genus is sister to the *Atropa* genus.

## 3. Discussion

This is the first report of the complete CP genome of *L. chinense*. Meanwhile, it has been the third species of the *Lycium* genus to have its CP genome fully sequenced and analyzed, with those of the other two (*L. barbarum* and *L. ruthenicum*) published recently [35,36]. Interestingly, the size of the *L. chinense* CP genome (155,756 bp) is the largest among the three *Lycium* species, with 180 bp longer than that of *L. barbarum* and 763 bp longer than that of *L. ruthenicum*. This is consistent with the hypothesis that IR contraction and expansion is the main reason explaining size differences between CP genomes [35,36,37,38].

Analysis of the gene content indicated that there are 82 protein-coding genes with a standard ATG start codon among the total of 85 protein-coding genes in the *L. chinense* CP genome. While the remaining three genes contain abnormal ACG (*psbL* and *ndhD*) and ACT (*rps12*) start codons. ACG and ACT are meaningless as start codons, whereas in general they are synonymous codons which can still encode threonine. This phenomenon has also been reported in the model plant *Nicotiana tabacum*, where the start codon for *psbL* and *ndhD* are ACG as well [39]. Furthermore, an ACT start codon for *rps12* is also present in the mitochondrial genome of tube-dwelling diatom *Berkeleya fennica* [40]. Therefore, we hypothesize that the start codons of some genes such as *rps12*, *psbL* and *ndhD* might be mutated during the evolution. RNA editing is a common phenomenon in plant CP genomes and causes modifying mutations, changes reading frames, and regulates the expression of CP genes [41]. The *ndhD* and *psbL* can be normally transcribed due to RNA editing, and this phenomenon has also been described in *Linum usitatissimum* L [42], tobacco [43], spinach [44], and *Ampelopsis brevipedunculata* [45]. Amino acid transitions caused by RNA editing in the *L. chinense* CP genome exhibited the highest frequency of conversion from serine to leucine (S to L), representing a change from a hydrophilic into a hydrophobic amino acid. This is consistent with the general characteristics of high-level plant CP RNA editing [46,47]. In protein structures, hydrophobic residues are usually enclosed in the center If a hydrophilic mutation occurs, the stability of the protein structure will tend to decrease. While in severe cases, correct folding could be hindered. Therefore, we hypothesize that the detected changes induced by RNA editing might promote the formation of hydrophobic cores, making the proteins more stable [48,49].

SSRs, also called microsatellites, which are widely distributed across genomes, refer to a group of tandem sequences [50]. SSRs play a key role in genomes and have been widely used in genetic and genomic studies because of their extreme variability within species [51,52,53,54]. The analysis results of SSRs in our study are consistent with the hypothesis that CP SSRs have a strong A/T bias, which is a common finding in many plant species [55,56,57]. The abundance of AT nucleotides in CP genomes might be related to the presence of SSRs, which is related to the stability of AT and GC nucleotides, to some extent [11]. The SSRs in the *L. chinense* CP genome could be used as molecular markers resource in future genetic identification studies.

With their great potential for application in studies of phylogenetics, evolution and molecular systematics, CP genomes have been widely used to solve phylogenetic issues in various land plants [16,57,58]. To identify the evolutionary position of *L. chinense* within the Solanaceae family, we conducted multiple sequence alignments using 13 other Solanaceae CP genome sequences (contain only one IR region). Two species from Labiatae family were chosen as outgroups. The results of phylogenetic analysis using two types of software both showed that *L. chinense* is the most closely related to the species of *L. barbarum*, which is identical in appearance to *L. chinense*. However, their therapeutic efficacies and active ingredients differ from each other [59,60]. Nevertheless, much more research has been carried out on *L. barbarum*, which is also officially listed in the Chinese Pharmacopoeia (2015 version), while only a few papers have been published reporting *L. chinense*. Overall, this study has generated abundant information valuable for the investigation of the genetic diversity of *L. chinense* and the Solanaceae family more broadly; and it will facilitate the use of CP genome sequences for species identification.

## 4. Materials and Methods

### 4.1. Plant Material and DNA Extraction

Fresh leaves of the *L. chinense* were obtained from the Medicinal Plant Garden of Guangzhou University of Chinese Medicine. Total genomic DNA (gDNA) was extracted from those leaves using a DNeasy Plant Mini Kit (Qiagen, German). A DNA sample of good integrity and with both optical density (OD) 260/280 and OD 260/230 ratio greater than 1.8 was submitted to subsequent experiments.

### 4.2. Chloroplast Genome Sequencing and Assembly

The sequencing library was constructed using this gDNA, after being ultrasonically sheered into 250 bp fragments, and then be submitted to Next-generation Sequencing on an Illumina HiSeq 2000 platform. NGS platform generated 6.82 G of raw sequencing data, then the raw data were filtered by removing low quality reads with a sliding window quality cutoff of Q20 using Trimmomatic [61]. Using the complete sequence of *A. belladonna* chloroplast genome as a reference, CP-like reads were extracted from those clean reads and then be assembled using the Abyss2.0 program [62], resulting in a complete chloroplast genome sequence of *L. chinense*. PCR amplification was performed to verify the four junction regions between the IR regions and the LSC/SSC region. 

### 4.3. Gene Annotation and Genome Structure

Gene annotation of the *L. chinense* CP genome was conducted using the online program GeSeq-Annotation of Organellar Genomes (https://chlorobox.mpimp-golm.mpg.de/geseq.html) [63] with default parameters, which annotates not only protein-coding genes, but also tRNAs and rRNAs. Genious Pro. (version 4.8.4) was used to correct the annotation result by adjusting the open reading frame of coding genes. Further examination and revision of the annotation information were manually performed with the assistance of the CLC Sequence Viewer (version 8). After that the complete chloroplast genome was submitted using the program Sequin and an accession number of MK040922 was assigned from the GenBank. The Organellar Genome DRAW (OGDRAW) program was used to draw the physical map with default parameters and subsequent manual editing. [64]. 

GC content of CDS regions as well as the distribution of codon usage was analyzed using the Molecular Evolutionary Genetics Analysis (MEGA 6.06) [65,66]. The online program Predictive RNA Editor for Plants (PREP) suite was used, with a cutoff value set at 0.8, to predict potential RNA editing sites in 35 genes of the CP genome of *L. chinense* using [67]. Furthermore, by using MISA [68], SSRs were detected with default parameters. The forward and inverted repeats were determined by setting the parameter of Hamming Distance to 3 and the Minimal Repeat Size parameter to 30 with the REPuter program (https://bibiserv2.cebitec.uni-bielefeld.de/reputer) [69]. 

### 4.4. Genome Comparison and Phylogenetic Analysis

In order to compare the CP genome of *L. chinense* with others of *A. belladonna* (NC_004561.1), *L. barbarum* (MH032560.1) and *L. ruthenicum* (NC_039651), the mVISTA program (http://genome.lbl.gov/vista/index.shtml) [70] in the Shuffle-LAGAN mode [71] was carried out using the annotation of *L. chinense* as the reference. 

In order to identify the phylogenetic position of *L. chinense* within the tubiflorae lineages, two kinds of phylogenetic analysis software, MEGA 6.06 and Bayesian evolutionary analysis by sample tree (BEAST). First, 16 complete CP genome sequences were downloaded from the GenBank of National Center for Biotechnology Information (NCBI) database. Those cp genomes underwent a sequence alignment by the Multiple Alignment using Fast Fourier Transform (MAFFT) program [72] (https://www.ebi.ac.uk/Tools/msa/mafft/). Then by using Mega 6.06 [65], maximum likelihood (ML) analysis of those 16 CP genomes was performed [65] to find the best models, which is GTR + G + I, and then construct ML tree, taking the CP genome sequences of *Salvia japonica, Pogostemon cablin* as the outgroup. Support was estimated through 1000 bootstrap replicates to assess the reliability of the phylogenetic tree. BEAST 2.0 analysis was conducted under the guidance of the online tutorial (http://www.beast2.org/tutorials/). The first step is to upload the sequence alignment document to the Bayesian Evolutionary Analysis Utility (BEAUti), which is a user-friendly program for setting the evolutionary model and options for the Markov chain Monte Carlo (MCMC) analysis. The second step is to actually run BEAST using the input file that contains the data, model and settings. The phylogenetic tree was constructed using the figtree (http://tree.bio.ed.ac.uk/software/figtree/).

## 5. Conclusions

The CP genome of *L. chinense* is 155,756 bp in length and has a relatively conserved genome structure as well as gene content. Forty-nine repeated sequences and 231 SSRs, which are informative sources for the development of new molecular markers, were determined and analyzed. The genome structure and composition are similar among the four species belonging to the Solanaceae family. Comparing the CP genome organization of different species helps us to more deeply understand the process of chloroplast evolution. The results of the phylogenetic analysis, which is conducted among 16 species, demonstrate that *L. chinense* has a close relationship with *L. barbarum.* In a word, this paper will promote the further investigation of *L. chinense*.

## Figures and Tables

**Figure 1 plants-08-00087-f001:**
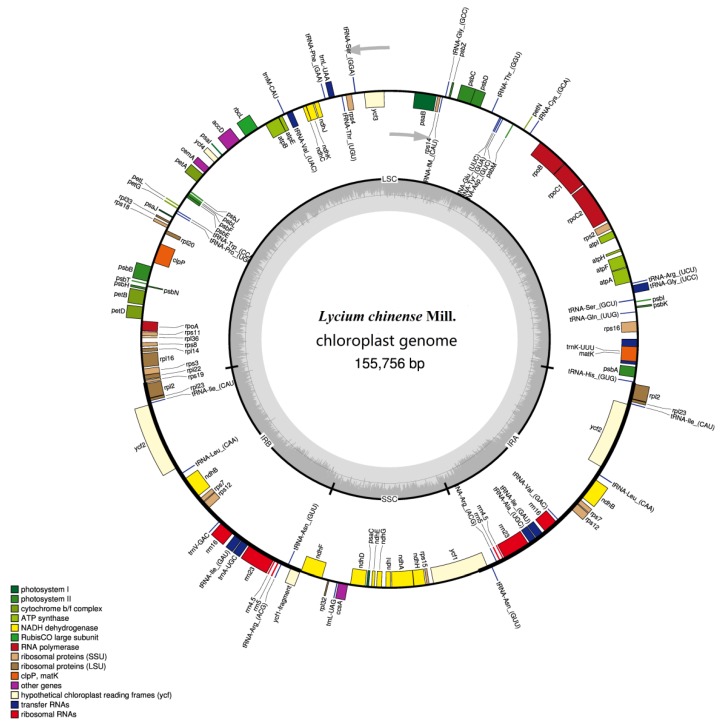
Gene map of the *L. chinense* chloroplast (CP) genome. Genes inside the circle are transcribed clockwise, and counterclockwise transcribed otherwise. Filled colors represent different functional groups according to the legend on the left bottom. The gray arrow represents gene direction. The darker gray in the inner circle corresponds to GC content, whereas the lighter gray corresponds to AT content.

**Figure 2 plants-08-00087-f002:**
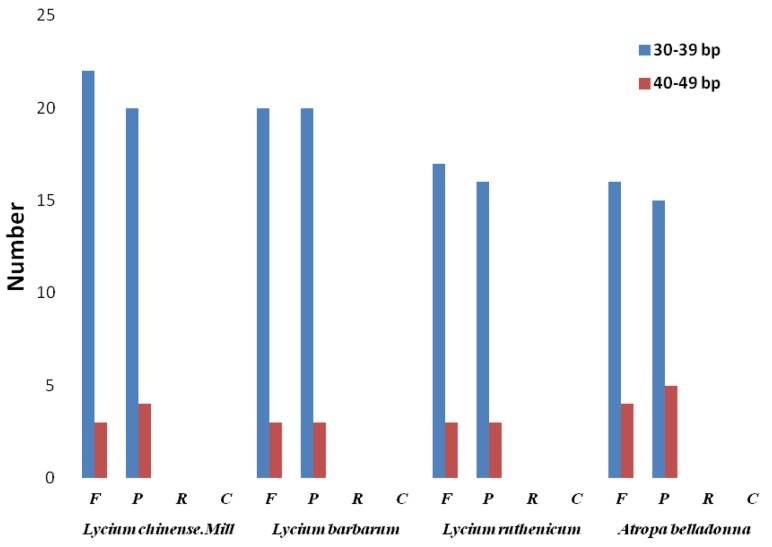
Repeat sequences in four chloroplast genomes. F, forward, P, palindrome, R, reverse, and C, complement, respectively. Colors represent repeats of different lengths, as indicated.

**Figure 3 plants-08-00087-f003:**
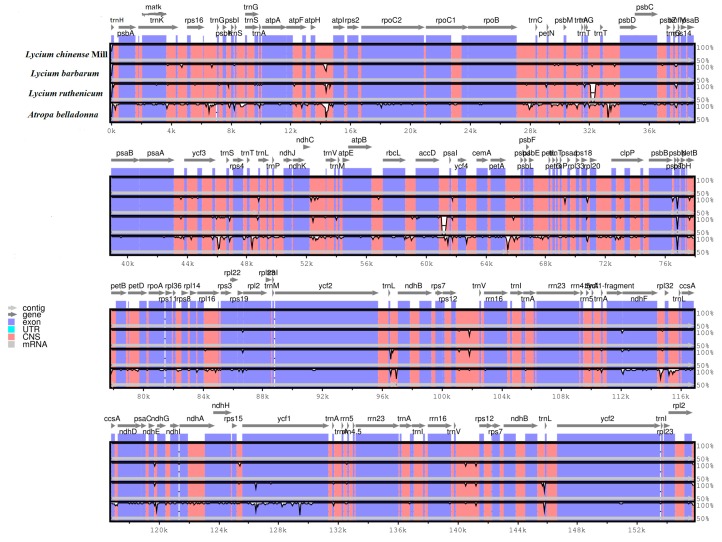
Sequence identity plot comparing five chloroplast genomes using mVISTA with *L. chinense* as the reference. Grey arrows and thick black lines above the alignment indicate genes with their orientation and the position of the inverted repeats (IRs), respectively. A cut-off of 70% identity was used for the plots, and the Y-scale represents the percentage identity ranging from 50% to 100%.

**Figure 4 plants-08-00087-f004:**
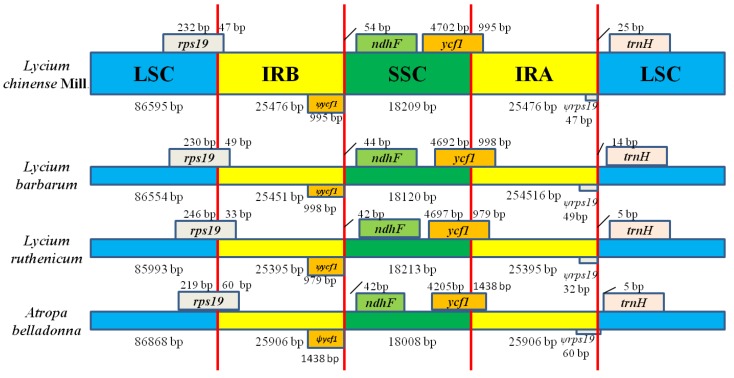
Comparison of the borders of large single copy regions (LSC), small single copy (LSC) and inverted repeat (IR) regions among four chloroplast genomes (not to scale). Number above the gene features means the distance between the ends of genes and the borders sites. The IRb/SSC border extended into the *ycf1* genes to create various lengths of *ycf1* pseudogenes among four chloroplast genomes, while the IRa/LSC border extended into the *rps19* genes to create various lengths of *rps19* pseudogenes.

**Figure 5 plants-08-00087-f005:**
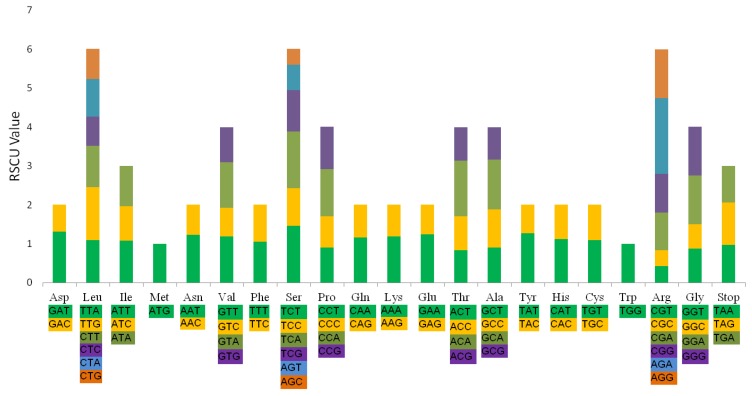
Codon content for 20 amino acids and stop codons in all protein-coding genes in the of *L. chinense* CP genome.

**Figure 6 plants-08-00087-f006:**
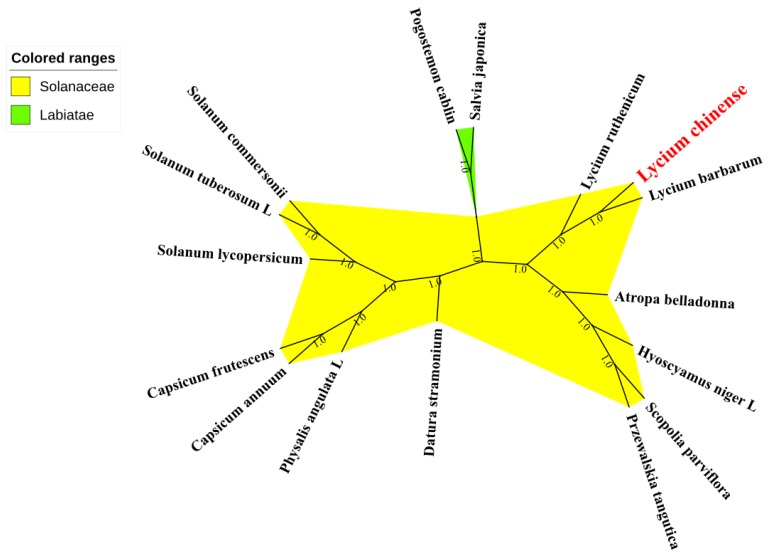
Phylogenetic relationships among 16 species inferred using maximum likelihood analyses of the complete chloroplast genome, excluding IRa regions. Numbers at nodes are values for bootstrap support. The position of *L. chinense* is indicated in red font. Sequences from *Salvia japonica* Thunb and *Pogostemon cablin* (Blanco) Ben were set as out group.

**Figure 7 plants-08-00087-f007:**
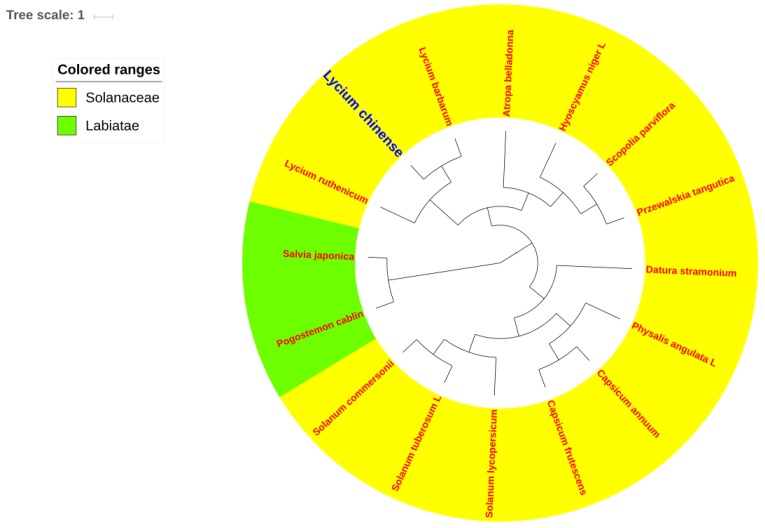
Phylogenetic relationships among 16 species inferred from Bayesian evolutionary analysis by sample tree (BEAST) analyses of complete chloroplast genome sequences, excluding IRa regions. The position of *L. chinense* is indicated in blue font. *Salvia japonica* Thunb and *Pogostemon cablin* (Blanco) Bent were used as outgroups.

**Table 1 plants-08-00087-t001:** Base composition in the chloroplast genome of *L. chinense.*

Region	Positions	T/U (%)	C (%)	A (%)	G (%)	Total (bp)
LSC		32.7	18.3	31.4	17.5	86,595
IRB		28.4	20.7	28.5	22.4	25,476
SSC		33.9	16.8	33.7	15.5	18,209
IRA		28.5	22.4	28.4	20.7	25,476
Total		31.5	19.2	30.7	18.6	155,756
CDS		31.3	17.9	30.5	20.3	79,700
	1st position	25.0	18.6	30.9	25.3	26,567
	2nd position	34.0	19.6	28.5	18.3	26,567
	3rd position	35.0	15.5	32.7	17.6	26,566

**Table 2 plants-08-00087-t002:** List of genes annotated in the chloroplast genomes of *L. chinense.*

Classification of Genes	Gene Names	Number
Photosystem I	*psaA*, *psaB*, *psaC*, *psaI*, *psaJ*	5
Photosystem II	*psbA*, *psbB*, *psbC*, *psbD*, *psbE*, *psbF*, *psbH*, *psbI*, *psbJ*, *psbK*, *psbL*, *psbM*, *psbN*, *psbT*, *psbZ*	15
Cytochrome b/f complex	*petA*, *petB **, *petD **, *petG*, *petL*, *petN*	6
ATP synthase	*atpA*, *atpB*, *atpE*, *atpF*, *atpH*, *atpI*	6
NADH dehydrogenase	*ndhA **, *ndhB* * (×2), *ndhC*, *ndhD*, *ndhE*, *ndhF*, *ndhG*, *ndhH*, *ndhI*, *ndhJ*, *ndhK*	12 (1)
RubisCO large subunit	*rbcL*	1
RNA polymerase	*rpoA*, *rpoB*, *rpoC1*, *rpoC2*	4
Ribosomal proteins (SSC)	*rps2*, *rps3*, *rps4*, *rps7* (×2), *rps8*, *rps11*, *rps12 *** (×2), *rps14*, *rps15*, *rps16 **, *rps18*, *rps19*	14 (2)
Ribosomal proteins (LSC)	*rpl2* (×2), *rpl14*, *rpl16*, *rpl20*, *rpl22*, *rpl23* (×2), *rpl32*, *rpl33*, *rpl36*	11
Ribosomal RNAs	*rrn 4.5* (×2), *rrn 5* (×2), *rrn 16* (×2), *rrn 23* (×2)	8 (4)
Protein of unkown function	*ycf1* (×2), *ycf2* (×2), *ycf3 ***, *ycf4*	6 (2)
Transfer RNAs	37 tRNAs (8 contain an intron, 7 in the inverted repeats region)	37 (7)
Other genes	*accD*, *ccsA*, *cemA*, *clpP*, *matK*	5
Total		130

* indicates gene containing one intron; while ** indicates gene with two introns.

**Table 3 plants-08-00087-t003:** Gene in the chloroplast genome of *L. chinense* with introns and exons.

Gene	Location	Exon I (bp)	Intron I (bp)	Exon II (bp)	Intron II (bp)	Exon III (bp)
*atpF*	LSC	410	704	145		
*clpP*	LSC	228	640	292	808	71
*ndhA*	SSC	539	1154	553		
*ndhB*	IR	777	679	756		
*petB*	LSC	6	750	642		
*petD*	LSC	8	742	475		
*rpl16*	LSC	396	1016	9		
*rpl2*	IR	434	666	391		
*rpoC1*	LSC	1616	737	430		
*rps12*	IR	26	536	232		
*rps16*	LSC	227	822	40		
*trnA-UGC*	IR	38	681	35		
*trnI-GAU*	IR	34	717	37		
*trnK-UUU*	LSC	36	2513	37		
*trnL-UAA*	LSC	35	497	50		
*trnV-UAC*	LSC	35	565	38		
*ycf3*	LSC	154	756	229	744	124

**Table 4 plants-08-00087-t004:** Types and numbers of simple sequence repeats (SSRs) in the *L. chinense* chloroplast genome.

SSR Type	Repeat Unit	Amount	Ratio (%)
Mono	A/T	106	99.1
	C/G	1	0.9
Di	AC/GT	1	2.2
	AG/CT	16	34.8
	AT/TA	29	63.0
Tri	AAC/GTT	9	13.4
	AAG/CTT	20	30.0
	AAT/ATT	21	31.3
	ACC/GGT	1	1.4
	ACG/CGT	1	1.4
	ACT/AGT	2	3.0
	AGC/CTG	5	7.5
	AGG/CCT	4	6.0
	ATC/ATG	4	6.0
Tetra	AAAC/GTTT	3	27.3
	AAAG/CTTT	1	9.1
	AAAT/ATTT	5	45.4
	AATC/ATTG	1	9.1
	AGAT/ATCT	1	9.1

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
