# Peer review of "Sequencing and Structural Analysis of the Complete Chloroplast Genome of the Medicinal Plant *Lycium"

_plants, 2019, doi:10.3390/plants8040087_

Reviewer 1 Report

Yang et al.  have sequenced the chloroplast genome of an important chinese herb Lycium chinense Mill. This study presents comprehensive analyses of the chloroplast genome sequence and might be of interest in the herbal medicine community. 

However, the authors' claim that this is the first chloroplast genome sequence in the Lycium genus is erroneous. Lycium barabarum and Lycium ruthenicum chloroplast genomes have been published in 2018. Therefore, authors need to include this information in their introduction, include these species in their comparative analyses + phylogenetic tree in the results and discuss the resulting implications in discussions and conclusions. 

The other major concern is that the abstract and contents of the manuscript are riddled with grammatical errors and in some places, even spelling errors. The authors need to engage a professional to go over and correct the language and grammar of the paper.

Minor comments:

Line 220-221: the authors need to to mention which filtering/trimming tool was used to clean the short reads and also cite the relevant paper.

Line 30: What is meant by 'functional food'? The citation [2] does not seem to be correct.

Overall, I would like the authors to resubmit for review after the major concerns are addressed. Then, the reviewers will be able to provide a more detailed feedback.

Author Response

Dear reviewer:

Thank you so much for your advice. I have revised the article according to your advices.

Point 1: However, the authors' claim that this is the first chloroplast genome sequence in the Lycium genus is erroneous. Lycium barabarum and Lycium barabarum chloroplast genomes have been published in 2018. Therefore, authors need to include this information in their introduction, include these species in their comparative analyses + phylogenetic tree in the results and discuss the resulting implications in discussions and conclusions.

Response 1: Thank you so much for your advice. I am so sorry for this, I haven't found any articles about the chloroplast genome of these two species, Lycium barabarum and Lycium ruthenicum chloroplast genomes. And according to your adviceI have included this information in our introduction, include these species in our comparative analyses and phylogenetic tree in the results and discuss the resulting implications in discussions and conclusions.

Point 2: The other major concern is that the abstract and contents of the manuscript are riddled with grammatical errors and in some places, even spelling errors. The authors need to engage a professional to go over and correct the language and grammar of the paper.

Response 2: Thank you so much for your advice. I am so sorry for this, And according to your adviceI have engage a professional to go over and correct the language and grammar of the paper.

Point 3: Line 220-221: the authors need to to mention which filtering/trimming tool was used to clean the short reads and also cite the relevant paper.

Response 3: Thank you so much for your advice. And according to your adviceI have mentioned which filtering/trimming tool was used to clean the short reads and also cite the relevant paper.

Point 4: Line 30: What is meant by 'functional food'? The citation [2] does not seem to be correct.

Response 4: Thank you so much for your advice. And according to your advice, I have corrected the citation [2]. And I think functional food refers to a food with a specific function, suitable for a specific group of people to eat, can adjust the function of the body, and not for the purpose of treatment. I have deleted this words from the manuscript, since It will produce ambiguity.

Reviewer 2 Report

In this study, the authors sequenced and assembled the chloroplast genome of Lycium chinense. As such it is yet another study where a chloroplast genome is exhaustively reported. It may potentially be published in Plants – eventually. At the moment, the main problem is language. This starts with typos and ends with wacky sentences. The authors need to use a language service or have a native English speaker revising the manuscript before it can be thoroughly evaluated.

From a technical view, most analyses are fine. Tables S1-S3 should be removed from the actual manuscript and Table 3 can be deleted. The main problem I see is the phylogenetic analysis, which is very shallow – ideally the authors perform a more thorough analysis where they estimate divergence times, e.g. using BEAST. Also, the best phylogenetic model should be estimated using the program ModelTest. Note that you report only 100 bootstraps in the figure and 1’000 in the methods.

Another major issue I see are the references, which seem to be randomly picked – lick citations 9 or 10 on page 2. Also, very few citations are used again in the discussion. Please provide a better review of the literature, focusing more on the general processes rather than citing many examples of “here they sequenced another CP genome”. The format of the references is not consistent across all of them.

Lastly, check the manuscript for inconsistencies in numbers – e.g. reporting 130 genes (line 80) and 114 (line 17).

Author Response

Dear reviewer:

Thank you so much for your advice. I have revised the article according to your advices.

Point 1: In this study, the authors sequenced and assembled the chloroplast genome of Lycium chinense. As such it is yet another study where a chloroplast genome is exhaustively reported. It may potentially be published in Plants – eventually. At the moment, the main problem is language. This starts with typos and ends with wacky sentences. The authors need to use a language service or have a native English speaker revising the manuscript before it can be thoroughly evaluated.

Response 1Thank you so much for your advice. I am so sorry for this, And according to your adviceI have engage a professional to go over and correct the language and grammar of the paper.

Point 2: From a technical view, most analyses are fine. Tables S1-S3 should be removed from the actual manuscript and Table 3 can be deleted. The main problem I see is the phylogenetic analysis, which is very shallow – ideally the authors perform a more thorough analysis where they estimate divergence times, e.g. using BEAST. Also, the best phylogenetic model should be estimated using the program ModelTest. Note that you report only 100 bootstraps in the figure and 1’000 in the methods.

Response 2Thank you so much for your advice. And according to your adviceI have studied the BEAST software, but I am sorry that I have not fully understood this software yet. I can only add the phylogenetic tree generated by this software into the article (Figure 7), which is exactly the same as the phylogenetic tree generated by the MEGA software(Figure 6). I hope you can forgive me for this. Tables S1-S3 are deleted, and I don't delete the Table 3, because I think it necessary the analyse it , please forgive me about this. And The 100 in the figure means 100%. And I have used the best model to do the phylogenetic analysis.

Point 3:Another major issue I see are the references, which seem to be randomly picked – lick citations 9 or 10 on page 2. Also, very few citations are used again in the discussion. Please provide a better review of the literature, focusing more on the general processes rather than citing many examples of “here they sequenced another CP genome”. The format of the references is not consistent across all of them.

Response 3Thank you so much for your advice. I am so sorry for this, And according to your adviceI have corrected the fomat of the referencescheck the references, and the discussion has been revised and add some references to the discussion.

Point 4: Lastly, check the manuscript for inconsistencies in numbers – e.g. reporting 130 genes (line 80) and 114 (line 17).

Response 4Thank you so much for your advice. But the number is right. 114 plus 16 equals 130.

Reviewer 3 Report

In this paper the authors announce the plastome sequence forLycium chinense, a species endemic to China that produces an edible fruit and is important in traditional Chinese herbology.   The science and the methodology seem to be OK but the paper is not ready for publication. Numerous grammatical, sentence construction, and word choice errors occur throughout the paper.  The authors should consider having a professor of English or a native English speaker review the text before they resubmit.  In my comments below, I have provided some changes regarding how the information is presented but also some English language usage edits.  This is not a complete list of edits, as it is out of the purview of a scientific reviewer to highly edit a paper for language usage to the degree that would be required to make this paper acceptable.

Major Changes

Introduction:

Line 28 – “at the magnificent” – the scientific primary literature shouldn’t include hyperbole in its descriptions of cultural traditions.  I recommend, “Lycium chinense plays an important role in Chinese traditional medicine.”

What makes this a “functional food”?

Line 36 – “among which Lycium is one of the most important.”  What makes this genus “important”? 

Lines 42-43 – These sentences contradict one another.  The authors site a study where barcodes were effectively used to identify relationships in Lyciumbut then say they are not effective.  Which is it?

Section 2.2

Beginning on line 97 there is a comparison of L. chinenseto other species.  These other species and their relationship toL. chinensehas not been introduced or described making it difficult for the reader to understand its importance.

Line 99 – what is meant by “forward and palindromic”

Section 2.5

The first paragraph, lines 146-154 is difficult to understand.  The second sentence is non-sensical.  I can’t tell if the difficulty I’m having is from the author’s English language usage or an incomplete description of the data.  It should be rewritten.

Section 2.6

The sentence on lines 172-173 doesn’t make sense to me.  It should be re-written.

Table S3

The Genus column is redundant.  It can be removed

3. Discussion

The discussion is shallow. The authors should expand on the meaning of the concepts and how they fit with other studies.

Line 195 – How can RNA editing change reading frames?

Line 211 – The final sentence is vague.  How will the existence of this sequence help with future studies?

4. Materials and Methods

In general this section lacks detail and needs to be edited.  As you consider how much detail to include ask yourself, “Could another scientist recreate my methodology?”

Suggested English language edits for the abstract and introduction.  I have included this as an example of the level of editing required for much of the manuscript.

Line 9 – Pharmaceutical is misspelled

Line 13 – …genome of L. chinensewas sequenced…

Line 18 – …have the standard…

Line 21 – L. chinense is a sister taxon to…

Line 21 – In general, the complete…

Line 23 – …species, and phylogenetic…

Line 32 – …the dry root bark of L. chinenseis widely used…

Line 35 – …belongs to the Solanaceae family

Line 36 – …24 genera

Line 36 – ...among which Lycium is one of the most important.

Line 39 – …uniquely located…

Line 41 – DNA barcoding has been…

Line 42 – …analyze the phylogenetic relationships within the…

Line 45 – … (CP) genomes have…

Line 46 - …land plants. Chloroplasts play an indispensable role in a plant cell’s utilization of photosynthesis, carbon fixation, and… (plastids perform many roles important for plant cell physiology.  I recommend the authors mention these)

Line 48 – …sequencing an entire genome is feasible for most laboratories…

Line 49 – According to published reports, the …

Line 50 – …are highly conserved with…

Lines 50-51 – You should explain what the IR and SC regions are.

Lines 51-53 – Chloroplast genomes can be used to study evolutionary relationships at a taxonomic level as a result of being maternally inherited, haploid, and highly conserved… 

Line 55 – Transcriptome analysis of L. chinense leaves have…

Lines 56-58 – Here we report the CP genome of L. chinense based on next-generation…

Line 60 – To the best of our knowledge, this is the first comprehensive analysis of the CP…

Author Response

Dear reviewer:

Thank you so much for your advice. I have revised the article according to your advices.

Point 1In this paper the authors announce the plastome sequence forLycium chinense, a species endemic to China that produces an edible fruit and is important in traditional Chinese herbology.   The science and the methodology seem to be OK but the paper is not ready for publication. Numerous grammatical, sentence construction, and word choice errors occur throughout the paper.  The authors should consider having a professor of English or a native English speaker review the text before they resubmit.  In my comments below, I have provided some changes regarding how the information is presented but also some English language usage edits.  This is not a complete list of edits, as it is out of the purview of a scientific reviewer to highly edit a paper for language usage to the degree that would be required to make this paper acceptable.

Response 1: Thank you so much for your advice. I am so sorry for this, And according to your adviceI have engage a professional to go over and correct the language and grammar of the paper.

Point 2Line 28 – “at the magnificent” – the scientific primary literature shouldn’t include hyperbole in its descriptions of cultural traditions.  I recommend, “Lycium chinense plays an important role in Chinese traditional medicine.”

Response 2: Thank you so much for your advice. And according to your adviceI have rewrite it.

Point 3What makes this a “functional food”?

Response 3: Thank you so much for your advice. I think functional food refers to a food with a specific function, suitable for a specific group of people to eat, can adjust the function of the body, and not for the purpose of treatment. I have deleted this words from the manuscript, since It will produce ambiguity.

Point 4Line 36 – “among which Lycium is one of the most important.”  What makes this genus “important”? 

Response 4: Thank you so much for your advice. According to the citation[7], the fruit, leaf, root bark, and young shoot of many species of the genus Lycium L. have long been used as local foods and/or medicines, so it is important.

Point 5Lines 42-43 – These sentences contradict one another.  The authors site a study where barcodes were effectively used to identify relationships in Lyciumbut then say they are not effective. Which is it?

Response 5: Thank you so much for your advice. I am so sorry, in this sentence my intended meaning is DNA barcoding was tried to identify and analysis the phylogenetic relationships of the Lycium genus, but it is not so effective. So it needs more effective methods.

Point 6Beginning on line 97 there is a comparison of L. chinenseto other species.  These other species and their relationship toL. chinensehas not been introduced or described making it difficult for the reader to understand its importance.

Response 6: Thank you so much for your advice. According to your advices, I introduced their relationship in the text, they belong to the same family.

Point 7Line 99 – what is meant by “forward and palindromic”

Response 7: Thank you so much for your advice. Forward means forward repeat sequencethe direction of this repeat sequence is the same and palindromic means palindromic repeat sequenceit is an inverted repeat in double-stranded DNA, and when the double strand of the sequence is opened, a hairpin structure can be formed. This sequence is called a palindrome sequence.

Point 8The first paragraph, lines 146-154 is difficult to understand.  The second sentence is non-sensical.  I can’t tell if the difficulty I’m having is from the author’s English language usage or an incomplete description of the data.  It should be rewritten.

Response 8: Thank you so much for your advice. According to your advices, it has been rewritten.

Point 9 The sentence on lines 172-173 doesn’t make sense to me.  It should be re-written.

Response 9: Thank you so much for your advice. According to your advices, it has been rewritten.

Point 10 Table S3 The Genus column is redundant.  It can be removed

Response 10: Thank you so much for your advice. According to your advices, it has been removed.

Point 11 The discussion is shallow. The authors should expand on the meaning of the concepts and how they fit with other studies.

Response 11: Thank you so much for your advice. And According to your advices, I have expand on the meaning of the concepts and how they fit with other studies.

Point 12 Line 195 – How can RNA editing change reading frames?

Response 12Thank you so much for your question. I want to explain thisRNA editing is one of the post-transcriptional regulation mechanisms of gene expression in the chloroplast of higher plants, so it can change reading frames.

Point 13 Line 211 – The final sentence is vague.  How will the existence of this sequence help with future studies?

Response 13Thank you so much for your question. I want to explain thisWhat I want to say in this sentence is that this study can promote the identification of future species using the chloroplast genome, rather than saying that the existence of this sequence can promote any research.

Point 14 In general this section(Methods and materials)lacks detail and needs to be edited.  As you consider how much detail to include ask yourself, “Could another scientist recreate my methodology?”

Response 14Thank you so much for your advice. According to your advices, I have added details to this part.

Point 15 Suggested English language edits for the abstract and introduction.  I have included this as an example of the level of editing required for much of the manuscript.

Response 15Thank you so much for your advice. According to your advices, I have edited the article.

Line 9 – Pharmaceutical is misspelled

Thank you so much for your advice. According to your advices, I have edited it.

Line 13 – …genome of L. chinensewas sequenced…

Thank you so much for your advice. According to your advices, I have edited it.

Line 18 – …have the standard…

Thank you so much for your advice. According to your advices, I have edited it.

Line 21 – L. chinense is a sister taxon to…

Thank you so much for your advice. According to your advices, I have edited it.

Line 21 – In general, the complete…

Thank you so much for your advice. According to your advices, I have edited it.

Line 23 – …species, and phylogenetic…

Thank you so much for your advice. According to your advices, I have edited it.

Line 32 – …the dry root bark of L. chinenseis widely used…

Thank you so much for your advice. According to your advices, I have edited it.

Line 35 – …belongs to the Solanaceae family

Thank you so much for your advice. According to your advices, I have edited it.

Line 36 – …24 genera

Thank you so much for your advice. According to your advices, I have edited it.

Line 36 – ...among which Lycium is one of the most important.

Thank you so much for your advice. According to your advices, I have edited it.

Line 39 – …uniquely located…

Thank you so much for your advice. According to your advices, I have edited it.

Line 41 – DNA barcoding has been…

Thank you so much for your advice. According to your advices, I have edited it.

Line 42 – …analyze the phylogenetic relationships within the…

Thank you so much for your advice. According to your advices, I have edited it.

Line 45 – … (CP) genomes have…

Thank you so much for your advice. According to your advices, I have edited it.

Line 46 - …land plants. Chloroplasts play an indispensable role in a plant cell’s utilization of photosynthesis, carbon fixation, and… (plastids perform many roles important for plant cell physiology.  I recommend the authors mention these)

Thank you so much for your advice. According to your advices, I have edited it.

Line 48 – …sequencing an entire genome is feasible for most laboratories…

Thank you so much for your advice. According to your advices, I have edited it.

Line 49 – According to published reports, the …

Thank you so much for your advice. According to your advices, I have edited it.

Line 50 – …are highly conserved with…

Thank you so much for your advice. According to your advices, I have edited it.

Lines 50-51 – You should explain what the IR and SC regions are.

Thank you so much for your advice. But I have explain it in the abstract, IR means inverted repeat region, SC means single copy region.

Lines 51-53 – Chloroplast genomes can be used to study evolutionary relationships at a taxonomic level as a result of being maternally inherited, haploid, and highly conserved…

Thank you so much for your advice. According to your advices, I have edited it.

Line 55 – Transcriptome analysis of L. chinense leaves have…

Thank you so much for your advice. According to your advices, I have edited it.

Lines 56-58 – Here we report the CP genome of L. chinense based on next-generation…

Thank you so much for your advice. According to your advices, I have edited it.

Line 60 – To the best of our knowledge, this is the first comprehensive analysis of the CP…

Thank you so much for your advice. According to your advices, I have edited it.

Round  2

Reviewer 1 Report

I am satisfied with the extensive changes made by the authors in response to my previous comments on the manuscript. I commend the authors on the fast turnaround and thorough rewriting and reanalysis that they have done, vastly improving the manuscript. 

Author Response

Response to Reviewer 1 Comments

Point : I am satisfied with the extensive changes made by the authors in response to my previous comments on the manuscript. I commend the authors on the fast turnaround and thorough rewriting and reanalysis that they have done, vastly improving the manuscript.

Response : Thanks so much for your confirmation of the changes I made. I will continue to work hard to write good articles. Thanks again and best wishes to you.

Reviewer 3 Report

The authors have significantly improved their manuscript from the original draft and I think it is now passable.  I recommend a few small changes in the text.

Line 19:  “codon” is misspelled

Lines 43-45:  This sentence is still difficult to read.  I recommend the following edit,  “There has been an attempt to define DNA barcode sequences to identify and analyze the phylogenetic relationships within the Lycium genus, however, this has been unsuccessful {Ni, 2016 #85}

Line 46 …facilitate the investigation…

Line 52 …comprised of two…

Line 53 …the two IRs separated…

Line 59 …genome is…

Line 60  sequencing is misspelled

Line 67 …consisting of large and small…

Line 74 This phenomenon, with is commonly…

Line 94  …are represented…

Line 96  …only one was a…

Line 114 …demonstrate that…

Line 133   There’s a semicolon followed by a comma.  Choose one or the other.

Line 243  A DNA sample of good…

Lines 249-250  …then the raw data were filtered by removing low quality…

Line 290  (remove ‘The’ from the beginning of the sentence and add ‘conducted’)  BEAST 2.0 analysis was conducted under…

Author Response

Response to Reviewer 3 Comments

Point 1The authors have significantly improved their manuscript from the original draft and I think it is now passable.  I recommend a few small changes in the text..

Response 1Thank you so much for your confirmation. And best wishes to you.

Point 2Line 19:  “codon” is misspelled

Response 2: Thank you so much for your advice. And according to your adviceI have corrected it.

Point 3This sentence is still difficult to read.  I recommend the following edit,  “There has been an attempt to define DNA barcode sequences to identify and analyze the phylogenetic relationships within the Lycium genus, however, this has been unsuccessful

Response 3: Thank you so much for your advice. And according to your adviceI have rewrited it 

Point 4Line 46 …facilitate the investigation…

Response4: Thank you so much for your advice. And according to your adviceI have rewrited it

Point 5Line 52 …comprised of two…

Response5: Thank you so much for your advice. And according to your adviceI have rewrited it

Point 6Line 53 …the two IRs separated.

Response6: Thank you so much for your advice. And according to your adviceI have rewrited it

Point 7Line 59 …genome is…

Response7Thank you so much for your advice. And according to your adviceI have rewrited it

Point 8Line 60  sequencing is misspelled

Response8Thank you so much for your advice. And according to your adviceI have corrected it

Point 9 Line 67 …consisting of large and small…

Response9Thank you so much for your advice. And according to your adviceI have rewrited it

Point 10 Line 94  …are represented…

Response10Thank you so much for your advice. According to your advices, I have rewrited it.

Point 11 Line 96  …only one was a…

Response11Thank you so much for your advice. According to your advices, I have rewrited it.

Point 12 Line 114 …demonstrate that…

Response12Thank you so much for your advice. According to your advices, I have rewrited it.

Point 13 Line 133   There’s a semicolon followed by a comma.  Choose one or the other.

Response13Thank you so much for your advice. According to your advices, I have rewrited it.

Point 14 Line 243  A DNA sample of good…

Response14Thank you so much for your advice. According to your advices, I have rewrited it.

Point 15 Lines 249-250  …then the raw data were filtered by removing low quality…

Response 15Thank you so much for your advice. According to your advices, I have rewrited it.

Point 16 Line 290  (remove ‘The’ from the beginning of the sentence and add ‘conducted’)  BEAST 2.0 analysis was conducted under

Response 16Thank you so much for your advice. According to your advices, I have rewrited it.